# *LSSVR* Model of G-L Mixed Noise-Characteristic with Its Applications

**DOI:** 10.3390/e22060629

**Published:** 2020-06-06

**Authors:** Shiguang Zhang, Ting Zhou, Lin Sun, Wei Wang, Baofang Chang

**Affiliations:** 1College of Computer and Information Engineering, Henan Normal University, Xinxiang 453007, China; sunlin@htu.edu.cn (L.S.); wangwei@htu.edu.cn (W.W.); changbaofang@htu.edu.cn (B.C.); 2School of Computer Science and Technology, Tianjin University, Tianjin 300350, China; 3Engineering Lab of Intelligence Business and Internet of Things, Xinxiang 453007, China; 4The State-Owned Assets Management Office, Henan Normal University, Xinxiang 453007, China

**Keywords:** Least square SVR, Gaussian–Laplacian mixed noise-characteristic, empirical risk loss, equality constraint, wind-speed forecasting

## Abstract

Due to the complexity of wind speed, it has been reported that mixed-noise models, constituted by multiple noise distributions, perform better than single-noise models. However, most existing regression models suppose that the noise distribution is single. Therefore, we study the Least square SVR of the Gaussian–Laplacian mixed homoscedastic (GLM−LSSVR) and heteroscedastic noise (GLMH−LSSVR) for complicated or unknown noise distributions. The ALM technique is used to solve model GLM−LSSVR. GLM−LSSVR is used to predict short-term wind speed with historical data. The prediction results indicate that the presented model is superior to the single-noise model, and has fine performance.

## 1. Introduction

In practical applications, if the data are collected in a multi-source environment, the noise distribution is complex and unknown. Therefore, it is almost impossible for a single-noise distribution to clearly describe the real-noise [1]. LSSVR is a method of LR that implements a sum-of-squares error function together with regularization, thus controlling the bias–variance trade-off [2,3]. It is intended to find the concealed linear structures in the original data [4,5]. For the sake of transition from linear to nonlinear function, the following generalization can be made [6]: by mapping input vectors into a high-dimensional feature space *H* (*H* is Hilbert space) through some nonlinear-mapping, seek the solution of the optimization problem in space *H*. Using a suitable kernel function K(•,•), nonlinear-mappings can be estimated by kernel LSSVR, which is an extended LR with kernel techniques. In recent years, LSSVR as a data-rich nonlinear forecasting tool has been increasingly welcomed [7], which is applicable in many different contexts [8,9,10], such as machine learning, optical character recognition, and especially wind speed/power forecasting.

Generally, the existing techniques used for wind-speed forecasting include: (i) physical; (ii) statistical (also called data-driven); and (iii) artificial intelligence (AI)-based methods. The physical models attempt to estimate wind flow around and inside the wind farm using physical laws governing the atmospheric behavior [11,12]. The statistical models seek the relationships between a set of explanatory variables and the on-line measured generation data, and the historical wind speed data recorded at the site are only used to establish the statistical model. We can model it in a variety of ways, including persistence method and auto-regressive model [13,14]. AI methods include artificial neural networks (ANNs) [15], deep learning [16], SVR machines [17,18], and the hybrid methods [19,20].

Suykens et al. [21,22,23] proposed least square support vector regression model with Gaussian noise (LSSVR, also known as kernel ridge regression (KRR)). Mixed-model based on multi-objective optimization [24,25], mixed-method based on singular spectrum analysis, firefly algorithm, and BP neural network predict wind speed with complicated noise [26], indicating that the mixed prediction method has the ability of powerful prediction. Mixed LSSVR machine [27] is applied to forecast the wind speed noise, which improves performance of wind-speed prediction. GLM−SVR [28] models fitted by Gaussian–Laplacian (G-L) mixed noise are developed, and good performance is obtained compared with the existing regression algorithm.

To solve the above problems, we study model LSSVR of G-L mixed noise-characteristic for complex or unknown noise distribution. In this case, we construct a technique to search the optimal solution of the corresponding regression task. Although many LSSVR algorithms have been implemented in past years, we exploit ALM method, as shown in Section 4. If the task is not differentiable or discontinuous, the sub gradient descent method can be employed, or the SMO [29] can also be used if there is a very large sample size.

The structure of this paper is as follows. Section 2 derives the optimal empirical risk loss by Bayesian principle. Section 3 constructs the LSSVR model of G-L mixed noise. Section 4 gives the solution and algorithm design of GLM−LSSVR. In Section 5, the numerical experiment of short-term wind-speed prediction is presented. Finally, we conclude the work.

## 2. Bayesian Principle to Mixed Noise Empirical Risk Loss

Given the Dataset
(1)$$D_{N}=\left\{\left(A_{1},y_{1}\right),\left(A_{2},y_{2}\right),\cdots ,\left(A_{N},y_{N}\right)\right\},$$
where $A_{i}={\left(x_{i1},x_{i2},\cdots ,x_{in}\right)}^{T}\in R^{n}$, $y_{i}\in R\left(i=1,2,\cdots ,N\right)$ is the training data. *R* represents real number set, $R^{n}$ is the *n*-dimensional Euclidean-Space, and *N* is the sample size. Superscript *T* is the transpose of matrix. Assuming that the sample of dataset $D_{N}$ is generated by the additive noise function ξ, the relationship between the measured value $y_{i}$ and predicted value $f(A_{i})$ is:(2)$$y_{i}=f\left(A_{i}\right)+\xi_{i},i=1,2,\cdots ,N$$
where $\xi_{i}$ is random, i.i.d. (independent, identical probability distribution) with $p(\xi_{i})$ of mean μ and standard deviation σ. Generally, the noise PDF (probability density function) p(ξ)=p(y−f(A)) is unknown. It is necessary to predict unknown target f(A) from training set $D_{f}\subseteq D_{N}$.

Following the authors of [30,31], the optimal empirical risk loss in the sense of Maximum Likelihood (MLE) is
(3)l(ξ)=l(A,y,f(A))=−Logp(y−f(A)),
i.e., the empirical risk loss l(ξ) is the log-likelihood of noise characteristic.

It is assumed that noise in Equation (Equation 2) is Laplacian, with PDF$p\left(\xi \right)=\frac{1}{2}e^{-|\xi |}$. By Equation (Equation 3), in MLE the optimal empirical risk loss should be l(ξ)=|ξ|.

Suppose noise in Equation (Equation 2) is Gaussian of zero mean and homoscedastic standard deviation σ. By Equation (Equation 3), the empirical risk loss of Gaussian noise with homoscedasticity is $l\left(\xi \right)=\frac{1}{2\sigma^{2}}\xi^{2}$. The noise in Equation (Equation 2) is Gaussian of zero mean and heteroscedastic standard deviation $\sigma_{i}$. By Equation (Equation 3), the empirical risk loss for Gaussian-noise with heteroscedasticity is $l\left(\xi_{i}\right)=\frac{1}{2\sigma_{i}^{2}}\xi_{i}^{2}$ (i=1,⋯,N).

Assume noise ξ in Equation (Equation 2) is the mixed noise of two kinds of noise with the PDFs $p_{1}\left(\xi \right)$ and $p_{2}\left(\xi \right)$, respectively. Suppose that $p\left(\xi \right)=\left[p_{1}{\left(\xi \right)}^{\lambda_{1}}\right]\cdot \left[p_{2}{\left(\xi \right)}^{\lambda_{2}}\right]$. By Equation (Equation 3), the corresponding empirical risk loss of mixed-noise is
(4)$$l\left(\xi \right)=\lambda_{1}\cdot l_{1}\left(\xi \right)+\lambda_{2}\cdot l_{2}\left(\xi \right).$$
where $l_{1}\left(\xi \right)>0,l_{2}\left(\xi \right)>0$ are the convex empirical risk losses of the above two kinds of noise characteristic, respectively. The weight factors are $\lambda_{1},\lambda_{2}\geq 0$ and $\lambda_{1}+\lambda_{2}=1$.

Figure 1 displays the Gaussian–Laplacian (G-L) empirical risk loss of different parameters (the parameter lambda is λ) [29].

## 3. LSSVR Model of G-L Mixed Noise-Characteristic

Given the training samples $D_{f}\subseteq D_{N}$, construct the linear regressor $f\left(A\right)=\varpi^{T}\cdot A+b$. To deal with nonlinear problems, it can be summarized as follows: mapping input vectors $A_{i}\in R^{n}$ into high-dimension feature space *H* through the nonlinear mapping Φ (take a prior distribution), induced by nonlinear kernel function $K(A_{i},A_{j})$, kernel mapping Φ is any positive definite Mercer kernel.

**Definition** **1**([6,28]). *Positive definite Mercer kernel: Assume that X is a subset of $R^{n}$. Assume that the kernel function $K(A_{i},A_{j})$ defined on X×X is a positive definite Mercer kernel functionl the kernel mapping* Φ *is called a positive definite Mercer kernel if there is mapping Φ:X→H (H is Hilbert Space), such that*
(5)$$K\left(A_{i},A_{j}\right)=\left(\Phi \left(A_{i}\right)\cdot \Phi \left(A_{j}\right)\right),\left(i,j=1,2,\cdots ,N\right).$$
*where (·) represents the inner-product in Space H.*


Therefore, the optimization problem of Space *H* is solved. At present, the input vectors $\left(A_{i}\cdot A_{j}\right)$ are replaced by inner product $\left(\Phi \left(A_{i}\right)\cdot \Phi \left(A_{j}\right)\right)$ in feature space *H*. Through the use of kernel $K\left(A_{i},A_{j}\right)=\left(\Phi \left(A_{i}\right)\cdot \Phi \left(A_{j}\right)\right)$, the linear model be extended to a nonlinear LSSVR.

In general, the mixed distribution has fine approximation ability to any continuous distribution. When there is no prior knowledge of real-noise, it can well adapt to unknown or complicated noise. Thus, it is presented that a uniform model LSSVR with mixed noise characteristics (M−LSSVR). The primal problem of model M−LSSVR is formalized as
(6)$$\begin{array}{c} Min\{g_{P_{M-LSSVR}}=\frac{1}{2}\varpi^{T}\cdot \varpi +\frac{C}{N}\cdot [\lambda_{1}\cdot \sum_{i=1}^{N}\left(l_{1}\left(\xi_{i}\right)\right)+\\ \;\;\;\;\;\;\lambda_{2}\cdot \sum_{i=1}^{N}\left(l_{2}\left(\xi_{i}\right)\right)]\}\\ s.t.:\;\xi_{i}=y_{i}-\varpi^{T}\cdot \Phi \left(A_{i}\right)-b \end{array}$$
where parameter $\varpi \in R^{n}$ represents weight-vector, *b* is the bias-term, C>0 is the penalty parameter, and the weight factors are $\lambda_{1},\lambda_{2}\geq 0$, $\lambda_{1}+\lambda_{2}=1$. $\left(A_{i},y_{i}\right)\in D_{N}$, Φ(A) is a nonlinear mapping which transfers the input dataset to a higher-dimensional feature space *H*. $\xi_{i}=y_{i}-\varpi^{T}\cdot \Phi \left(A_{i}\right)-b$ is the random noise variable at time i(i=1,2,⋯,N). $l_{1}\left(\xi_{i}\right)>0,l_{2}\left(\xi_{i}\right)>0\left(i=1,2,\cdots ,l\right)$ is the convex loss-functions for noise characteristic in sample-point $\left(A_{i},y_{i}\right)\in D_{N}$ ((i,j=1,2,⋯,N)).

In the application domain, most distributions do not obey Gaussian distribution, and they also do not satisfy Laplacian distribution. the noise distribution is complicated, and it is almost impossible to describe real noise with a single distribution. It has been reported that mixed noise models, constituted by multiple noise distributions, perform better than single-noise model [1]. As the function fitting -machine, the goal is to estimate an unknown function f(A) from dataset $D_{f}\subseteq D_{N}$. In this section, G-L mixed homoscedastic and heteroscedastic noise distributions are used to fit complicated noise characteristic.

### 3.1. LSSVR Model of G-L Mixed Homoscedastic Noise-Characteristic

Suppose noise in Equation (Equation 2) is Gaussian of zero mean and homoscedastic standard deviation σ. By Equation (Equation 3), we have that the empirical risk loss of homoscedastic-Gaussian-noise characteristic is $l_{1}\left(\xi \right)=\frac{1}{2\sigma^{2}}\cdot \xi^{2}$. The Laplacian-noise is $l_{2}\left(\xi \right)=\left|\xi \right|$. Adopting G-L mixed homoscedastic noise distribution to fit complicated noise-characteristic, by Equation (Equation 4), the empirical risk loss about G-L mixed homoscedastic noise is $l\left(\xi \right)=\frac{\lambda_{1}}{2\sigma^{2}}\cdot \xi^{2}+\lambda_{2}\cdot \left|\xi \right|$. Putting forward the LSSVR model of G-L mixed homoscedastic noise-characteristic (GLM−LSSVR), the primal problem of GLM−LSSVR is depicted as
(7)$$\begin{array}{c} Min\{g_{P_{GLM-LSSVR}}=\frac{1}{2}\varpi^{T}\cdot \varpi +\frac{C}{N}\cdot (\frac{\lambda_{1}}{2\sigma^{2}}\cdot \sum_{i=1}^{N}\xi_{i}^{2}\\ \;\;\;\;\;\;\;\;+\lambda_{2}\cdot \sum_{i=1}^{N}\xi_{i})\}\\ s.t.:\;\xi_{i}=y_{i}-\varpi^{T}\cdot \Phi \left(A_{i}\right)-b \end{array}$$
where parameter vector $\varpi \in R^{n}$, $\sigma^{2}$ is homoscedastic, C>0 is a penalty parameter, and the weight factors are $\lambda_{1},\lambda_{2}\geq 0$ and $\lambda_{1}+\lambda_{2}=1$.

**Proposition** **1.**
*The solution of the primal problem in Equation (Equation 7) of GLM−LSSVR is existent and unique about ϖ.*


**Theorem** **1.**
*The dual problem of the primal problem in Equation (Equation 7) is*
(8)$$\begin{array}{c} Max\{g_{D_{GLM-LSSVR}}=-\frac{1}{2}\sum_{i=1}^{N}\sum_{j=1}^{N}\alpha_{i}\cdot \alpha_{j}\cdot K\left(A_{i},A_{j}\right)+\\ \;\;\;\;\;\;\sum_{i=1}^{N}\alpha_{i}\cdot y_{i}-\frac{N}{2C\cdot \lambda_{1}}\sum_{i=1}^{N}{\left(\sigma^{2}\cdot \alpha_{i}-C\cdot \lambda_{2}\right)}^{2}\}\\ s.t.:\;\sum_{i=1}^{N}\alpha_{i}=0 \end{array}$$
*where $\sigma^{2}$ is homoscedastic, C>0 is a penalty parameter, and the weight factors are $\lambda_{1},\lambda_{2}\geq 0$ and $\lambda_{1}+\lambda_{2}=1$.*


**Proof.** We introduce Lagrange functional L(ϖ,b,α,ξ) as $L\left(\varpi ,b,\alpha ,\xi \right)=\frac{1}{2}\varpi^{T}\cdot \varpi +\frac{C}{N}\cdot \left(\frac{\lambda_{1}}{2\sigma^{2}}\cdot \sum_{i=1}^{N}\xi_{i}^{2}+\lambda_{2}\cdot \sum_{i=1}^{N}\xi_{i}\right)+\sum_{i=1}^{N}\alpha_{i}\left(y_{i}-\varpi^{T}\cdot \Phi \left(A_{i}\right)-b-\xi_{i}\right).$Minimizing L(ϖ,b,α,ξ) and deriving the partial-derivative ϖ,b,ξ, respectively, on the basis of KKT-conditions, we get
$$\nabla_{\varpi }\left(L\right)=0,\nabla_{b}\left(L\right)=0,\nabla_{\xi }\left(L\right)=0.$$We obtain
$$\varpi =\sum_{i=1}^{N}\alpha_{i}\cdot \Phi \left(A_{i}\right),$$
$$\sum_{i=1}^{N}\alpha_{i}=0,$$
$$\frac{C}{N}\cdot \left(\frac{\lambda_{1}}{\sigma^{2}}\cdot \xi_{i}+\lambda_{2}\right)-\alpha_{i}=0\left(i=1,2,\cdots ,N\right).$$The extreme condition is replaced by L(ϖ,b,α,ξ), and the maximum value of α is obtained. The dual problem in Equation (Equation 8) of the primal problem in Equation (Equation 7) is derived. □

Therefore,
$$\varpi_{i}=\sum_{i=1}^{N}\alpha_{i}\cdot \Phi \left(A_{i}\right),$$
$$b=\frac{1}{N}\sum_{i=1}^{N}\left[y_{i}-\sum_{j=1}^{N}\alpha_{i}\cdot K\left(A_{i},A_{j}\right)-\frac{1}{\lambda_{1}}\cdot \left(\frac{N\cdot \sigma^{2}\cdot \alpha_{i}}{C}-\lambda_{2}\right)\right].$$

The decision-maker for GLM−LSSVR may be represented as
$$f\left(A\right)=\varpi^{T}\cdot \Phi \left(A\right)+b=\sum_{i=1}^{N}\alpha_{i}K\left(A_{i},A\right)+b.$$
where the parameter vector $\varpi \in R^{n}$, $\Phi :R^{n}\to H$, $\left(\Phi \left(A_{i}\right)\cdot \Phi \left(A_{j}\right)\right)$ is the inner-product of *H* and $K\left(A_{i},A_{j}\right)=\left(\Phi \left(A_{i}\right)\cdot \Phi \left(A_{j}\right)\right)$ is the kernel-function.

Suppose the noise in Equation (Equation 2) is Gaussian homoscedastic noise, which is Gaussian noise of zero mean and the homoscedastic variance $\sigma^{2}$. Thus, the dual problem of LSSVR can be derived by Theorem 2:(9)$$\begin{array}{c} Max\{g_{D_{LSSVR}}=-\frac{1}{2}\sum_{i=1}^{N}\sum_{j=1}^{N}\left(\alpha_{i}\cdot \alpha_{j}\cdot K\left(A_{i},A_{j}\right)\right)\\ \;\;\;\;\;\;+\sum_{i=1}^{N}\left(\alpha_{i}\cdot y_{i}\right)-\frac{N}{2C}\cdot \sum_{i=1}^{N}\left(\sigma^{2}\cdot \alpha_{i}^{2}\right)\}\\ D_{GN-KRR}:s.t.\;\sum_{i=1}^{N}\alpha_{i}=0. \end{array}$$

### 3.2. LSSVR Model of G-L Mixed Heteroscedastic Noise-Characteristic

It is assumed that the noise in Equation (Equation 2) is Gaussian of zero mean and heteroscedastic standard deviation $\sigma_{i}$, that is $\sigma_{i}\neq \sigma_{j}$, i≠j(i,j=1,⋯,N). From Equation (Equation 3), the empirical risk loss of heteroscedastic Gaussian-noise characteristic is $l_{1}\left(\xi_{i}\right)=\frac{1}{2\sigma_{i}^{2}}\cdot \xi_{i}^{2}$ and the loss-function of Laplacian-noise is $l_{2}\left(\xi_{i}\right)=\left|\xi_{i}\right|$, (i=1,⋯,N). Utilizing G-L mixed heteroscedastic noise distribution to predict complicated noise-characteristic, from Equation (Equation 4), the loss function corresponding to G-L mixed heteroscedastic noise is $l\left(\xi_{i}\right)=\frac{\lambda_{1}}{2\sigma_{i}^{2}}\cdot \xi_{i}^{2}+\lambda_{2}\cdot \left|\xi_{i}\right|\left(i=1,\cdots ,N\right)$. The new model LSSVR with G-L mixed heteroscedastic noise-characteristic (GLMH−LSSVR) is proposed. The primal problem of GLMH−LSSVR is depicted as
(10)$$\begin{array}{c} Min\{g_{P_{GLMH-LSSVR}}=\frac{1}{2}\varpi^{T}\cdot \varpi +\\ \;\;\;\;\;\frac{C}{N}\cdot \left(\frac{\lambda_{1}}{2}\cdot \sum_{i=1}^{N}\frac{1}{\sigma_{i}^{2}}\xi_{i}^{2}+\lambda_{2}\cdot \sum_{i=1}^{N}\xi_{i}\right)\}\\ s.t.:\;\xi_{i}=y_{i}-\varpi^{T}\cdot \Phi \left(A_{i}\right)-b \end{array}$$
where the parameter vector is $\varpi \in R^{n}$, $\sigma_{i}^{2}\left(i=1,2,\cdots ,N\right)$ are heteroscedastic, and C>0 is the penalty parameter. The weight-factors are $\lambda_{1},\lambda_{2}\geq 0$ and $\lambda_{1}+\lambda_{2}=1$.

**Proposition** **2.**
*The solution of the primal problem in Equation (Equation 10) of GLMH−LSSVR is existent and unique about ϖ.*


**Theorem** **2.**
*The dual problem of model GLMH−LSSVR in Equation (Equation 10) is*
(11)$$\begin{array}{c} Max\{g_{D_{GLMH-LSSVR}}=-\frac{1}{2}\sum_{i=1}^{N}\sum_{j=1}^{N}\alpha_{i}\cdot \alpha_{j}\cdot K\left(A_{i},A_{j}\right)+\\ \;\;\;\sum_{i=1}^{N}\alpha_{i}\cdot y_{i}-\frac{N}{2C\cdot \lambda_{1}}\sum_{i=1}^{N}{\left(\sigma_{i}^{2}\cdot \alpha_{i}-C\cdot \lambda_{2}\right)}^{2}\}\\ s.t.:\;\;\sum_{i=1}^{N}\alpha_{i}=0 \end{array}$$
*where $\sigma_{i}^{2}\left(i=1,2,\cdots ,N\right)$ are heteroscedastic and C>0 is the penalty parameter. The weight factors are $\lambda_{1},\lambda_{2}\geq 0$ and $\lambda_{1}+\lambda_{2}=1$.*


**Proof.** It is easier to derive the proof of Theorem 2 by analogy with Theorem 1. □

We have
$$\varpi_{i}=\sum_{i=1}^{N}\alpha_{i}\cdot \Phi \left(A_{i}\right),$$
$$b=\frac{1}{N}\sum_{i=1}^{N}\left[y_{i}-\sum_{j=1}^{N}\alpha_{i}\cdot K\left(A_{i},A_{j}\right)-\frac{1}{\lambda_{1}}\cdot \left(\frac{N\cdot \sigma_{i}^{2}\cdot \alpha_{i}}{C}-\lambda_{2}\right)\right].$$

The decision-maker for GLMH−LSSVR may be expressed as
$$f\left(A\right)=\varpi^{T}\cdot \Phi \left(A\right)+b=\sum_{i=1}^{N}\alpha_{i}K\left(A_{i},A\right)+b,$$
where the parameter vector is $\varpi \in R^{n}$, $\Phi :R^{n}\to H$, and $K(A_{i},A_{j})$ is the kernel function.

Suppose noise in Equation (Equation 2) is G-L mixed-homoscedastic-noise, in which Gaussian-noise of zero mean and homoscedastic-variance $\sigma^{2}$, Theorem 1 can be deduced from Theorem 2.

## 4. Solution from ALM

In this section, we use Augmented Lagrange-multiplier method (ALM) [32] to solve the dual problem in Equation (Equation 8) by applying Gradient descent or Newton’s method to a sequence of equality-constrained problems. By eliminating equality constraints, arbitrary equality constraints can be reduced to equivalent unconstrained problems [33,34]. If there are large-scale training samples, some rapid optimization techniques can be combined with the proposed model, for example the sequential minimal optimization (SMO) algorithm [29] and the stochastic gradient decent (SDG) algorithm [35].

Theorems 1 and 2 provide effective recognition techniques for GLM−LSSVR and GLMH−LSSVR, respectively. In this section, we derive the solution from ALM and the algorithm for model LSSVR of G-L mixed homoscedastic noise characteristic (GLM−LSSVR). Analogously, the solution of model GLMH−LSSVR can be obtained by ALM method.

(1) Let dataset be $D_{N}=\left\{\left(A_{1},y_{1}\right),\left(A_{2},y_{2}\right),\ldots ,\left(A_{N},y_{N}\right)\right\}$, where $A_{i}\in R^{n}$, $y_{i}\in R$, i=1,…,N.

(2) The optimal parameters $C,\lambda_{1},\lambda_{2}$ were searched by using the 10-fold cross-validation strategy, and the appropriate kernel function K(•,•) was selected.

(3) Solve model GLM−LSSVR of the problem in Equation (Equation 8), and get the optimal solution $\alpha =(\alpha_{1},\cdots ,\alpha_{N})$.

(4) Build the decision-function as follows
$$f\left(A\right)=\varpi^{T}\cdot \Phi \left(A\right)+b=\sum_{i=1}^{N}\alpha_{i}K\left(A_{i},A\right)+b.$$

The parameter vector is $\varpi \in R^{n}$, $b=\frac{1}{N}\sum_{i=1}^{N}\left[y_{i}-\sum_{j=1}^{N}\alpha_{i}\cdot K\left(A_{i},A_{j}\right)-\frac{1}{\lambda_{1}}\cdot \left(\frac{N\cdot \sigma^{2}\cdot \alpha_{i}}{C}-\lambda_{2}\right)\right]$, $\Phi :R^{n}\to H$, $\left(\Phi \left(A_{i}\right)\cdot \Phi \left(A_{j}\right)\right)$ ((i,j=1,2,⋯,N)) is the inner product in *H*, $K\left(A_{i},A_{j}\right)=\left(\Phi \left(A_{i}\right)\cdot \Phi \left(A_{j}\right)\right)$ is a kernel function.

## 5. Case Study

This section tests and verifies the validity of constructed model GLM−LSSVR by comparing it with other techniques in the Heilongjiang, China dataset $D_{N}$. This case study consists of the following subsections: G-L mixed-noise characteristic of wind speed, prediction performance evaluation criteria, and short-term wind-speed forecasting based on an actual dataset.

### 5.1. G-L Mixed-Noise-Characteristic of Wind-Speed

To demonstrate the effectiveness of the proposed model, we collected wind speed data from Heilongjiang. The dataset consists of more than one year of wind speed data, recording wind speed values every 10 min. We first discovered the G-L mixed noise and conducted experiments on it. We found that turbulence is the main reason for the high uncertainty of wind speed random fluctuations. From the perspective of wind energy, the most significant feature of wind energy resources is their variability. Now, it shows the distribution of wind speed. Take a wind speed value every 5 s and calculate the histogram of wind speed within 1–2 h. Two typical distributions are given: one is calculated when the wind speed is high and the other is calculated when the wind speed is low (see Figure 2 and Figure 3, respectively).

We analyzed the one-month time-series dataset, and used the persistence method to investigate the error distribution [32]. The results show that the wind speed error ξ obtained from the persistence prediction is not subject to single distribution, while approximately to G-L mixed distribution, and PDF of ξ is $p\left(\xi \right)=\frac{1}{2}e^{-|\xi |}\cdot \frac{1}{2\sigma^{2}}\xi^{2}$, as shown in Figure 4.

As can be seen from the above charts and figures, wind speed error approximately satisfies G-L mixed distribution. This is a mixed kind of task.

### 5.2. Prediction Performance Evaluation Criteria

It is generally known that no prediction model forecasts perfectly. The predictable performance of ν−SVR, GN−SVR, LSSVR, and GLM−LSSVR also has certain evaluation criteria, for example MAE (mean absolute error), RMSE (root mean square error), MAPE (mean absolute percentage error), and SEP (the standard error of prediction). The four criteria be defined as follows:(12)$$MAE=\frac{1}{N}\sum_{i=1}^{N}\left|y_{i}^{\prime }-y_{i}\right|,$$
(13)$$MAPE=\frac{1}{N}\sum_{i=1}^{N}\frac{|y_{i}^{\prime }-y_{i}|}{y_{i}}\times 100\%,$$
(14)$$RMSE=\sqrt{\frac{1}{N}\sum_{i=1}^{N}{\left(y_{i}^{\prime }-y_{i}\right)}^{2}},$$
(15)$$SEP=\frac{RMSE}{\bar{y}}\times 100\%,$$
where *N* is the size of the dataset $D_{N}$, $y_{i}$ is the *i*th actual observed data, and $y_{i}^{\prime }$ is the *i*th forecasted-result. $\bar{y}$ is the mean value of observations $y_{i}\in D_{N}$[36,37,38,39,40]. MAE shows how similar the predicted value is to the observed value, while RMSE measures overall deviation between predicted value and observed value. MAPE is the ratio between error and observed value. SEP is the ratio of RMSE to average observation. They are dimensionless measurements of accuracy of wind speed system, and are sensitive to small changes.

### 5.3. Short-Term Wind-Speed Forecasting with Real dataset

In this section, 2160 consecutive data (1–2160, time span of 15-days) are extracted as the training set and 720 consecutive data (2161–2880, time span of 5-days) are extracted as the testing set. The input vector is **$\bar{A_{i}}=\left(x_{i-11},x_{i-10},\cdots ,x_{i-1},x_{i}\right)$**, $x_{j}$ is the actual observed data of wind speed at moment j(j=i−11,i−10,⋯,i), and the forecasting value is $x_{i+step}$, where step=1,3,6. That is, the above models are used to forecast wind speed of each point $x_{i}$ after 10, 30 and 60 min, respectively. Figure 5, Figure 6, Figure 7, Figure 8, Figure 9, Figure 10, Figure 11, Figure 12 and Figure 13 describe the forecasting results given by models ν−SVR, GN−SVR, LSSVR, and GLM−LSSVR.

The models ν−SVR, GN−SVR, LSSVR, and GLM−LSSVR were implemented in Matlab 7.8. Initial parameters of GLM−LSSVR were C∈[1,200], ν∈(0,1), and $\lambda_{1},\lambda_{2}\in \left[0,1\right]$. The optimal parameters $C,\nu ,\lambda_{1},\lambda_{2}$ were searched by using 10-fold cross-validation technique. The technology of parameter selection is studied in detail in [41,42]. In this simulation, parameters were set to $C=181,\nu =0.5,\lambda_{1}=0.5,\lambda_{2}=0.5$. The practical application demonstrates that both polynomial kernel and Gaussian kernel perform well under the assumption of smoothness. Under these circumstances, models ν−SVR, GN−SVR, LSSVR, and GLM−LSSVR employ polynomial and Gaussian kernel functions [43]:$$K\left(A_{i},A_{j}\right)={\left(\left(A_{i},A_{j}\right)+1\right)}^{d},$$
$$K\left(A_{i},A_{j}\right)=e^{-\frac{∥A_{i}-A_{j}{∥}^{2}}{\sigma^{2}}},$$
where *d* is a positive integer and σ is a positive number.

The dual problem of ν−SVR and SVR of the Gaussian-noise model (GN−SVR) and LSSVR are as follows.

ν−SVR: The authors of [41,44] define the dual problem of ν−SVR as
(16)$$\begin{array}{c} Max\{g_{D_{\nu -SVR}}=-\frac{1}{2}\sum_{i\in RSV}\sum_{j\in RSV}\left(\alpha_{i}^{*}-\alpha_{i}\right)\left(\alpha_{j}^{*}-\alpha_{j}\right)\\ \;\;\;\;\;\;\;\cdot K\left(A_{i},A_{j}\right)+\sum_{i\in RSV}\left(\alpha_{i}^{*}-\alpha_{i}\right)\cdot y_{i}\}\\ s.t.:\;\;\sum_{i=1}^{N}\left(\alpha_{i}^{*}-\alpha_{i}\right)=0\\ \;\;\;\;\;\;0\leq \alpha_{i}^{\left(*\right)}\leq \frac{C}{N},i=1,\cdots ,N\\ \;\;\;\;\;\;\sum_{i=1}^{N}\left(\alpha_{i}+\alpha_{i}^{*}\right)\leq C\cdot \nu ,i=1,\cdots ,N. \end{array}$$

GN−SVR: The authors of [45,46] studied SVR with equality constraints and inequality constraints. The loss-function of Gaussian-noise is $c\left(\xi_{i}\right)=\xi_{i}^{2}/2$, (i=1,⋯,N). Thus, thus dual problem of GN−SVR is
(17)$$\begin{array}{c} max\{g_{D_{GN-SVR}}=-\frac{1}{2}\sum_{i\in RSV}\sum_{j\in RSV}\left(\alpha_{i}^{*}-\alpha_{i}\right)\left(\alpha_{j}^{*}-\alpha_{j}\right)K\left(A_{i},A_{j}\right)+\\ \;\;\;\;\;\;\sum_{i\in RSV}\left(\alpha_{i}^{*}-\alpha_{i}\right)y_{i}-\frac{N}{2C}\sum_{i=1}^{N}\left(\alpha_{i}^{2}+{\left(\alpha_{i}^{*}\right)}^{2}\right)\}\\ s.t.:\;\;\sum_{i=1}^{N}\left(\alpha_{i}^{*}-\alpha_{i}\right)=0\\ \;\;\;\;\;\;0\leq \alpha_{i}^{\left(*\right)}\leq \frac{C}{N},i=1,\cdots ,N\\ \;\;\;\;\;\;\sum_{i=1}^{N}\left(\alpha_{i}+\alpha_{i}^{*}\right)\leq C\cdot \nu ,i=1,\cdots ,N. \end{array}$$

LSSVR: [22] studied LS−SVR for Gaussian-noise model. The dual problem of LS−SVR is
(18)$$\begin{array}{c} max\{g_{D_{LS-SVR}}=-\frac{1}{2}\sum_{i=1}^{N}\sum_{j=1}^{N}\left(\alpha_{i}\cdot \alpha_{j}\cdot K\left(A_{i},A_{j}\right)\right)\\ \;\;\;\;\;\;+\sum_{i=1}^{N}\left(\alpha_{i}\cdot y_{i}\right)-\frac{N}{2C}\cdot \sum_{i=1}^{N}\left(\alpha_{i}^{2}\right)\}\\ s.t.:\;\;\sum_{i=1}^{N}\alpha_{i}=0. \end{array}$$
where $\xi_{i},\xi_{i}^{*}$ are slack-variables. C>0, ν∈(0,1] are constants. For ν−SVR and GN−SVR, the size of ϵ is not gained, but is a variable whose value is compromised by a constant with the model complexity and relaxation variables through ν [35].

In Figure 5, Figure 8 and Figure 11, wind-speed forecasting-results at $A_{i}$-point of ν−SVR, GN−SVR, LSSVR, and GLM−LSSVR are presented after 10, 30, and 60 min, respectively. Figure 6, Figure 9, and Figure 12 show the error statistic of wind-speed prediction using the above four models. The box plots (Figure 7, Figure 10, and Figure 13) of several noise levels further intuitively demonstrate the comparative effect of error statistics using the above four wind-speed forecasting models. The statistical criteria of MAE, MAPE, RMSE and SEP are displayed in Table 1, Table 2 and Table 3.

From box-whisker plots in Figure 7, Figure 10, and Figure 13, as well as Table 1, Table 2 and Table 3, it can be concluded that, in most cases, the forecasting-error of GLM−LSSVR calculation is superior to ν−SVR, GN−SVR and LSSVR. With the increase of prediction horizon to 30 and 60 min, the forecasting error of different models increases and the relative error decreases. Thus, in these cases, it is not that important. However, Table 1, Table 2 and Table 3 show that, under all the criteria of MAE, MAPE, RMSE, and SEP, the Gaussian–Laplacian mixed-noise model is slightly better than the classical model.

## 6. Conclusions

Most existing regression-techniques suppose that the noise model is single. Wind-speed forecasting is complicated due to its volatility and uncertainty, thus it is difficult to model with a single-noise distribution. This section summarizes our main work: (1) optimal empirical risk loss of G-L mixed noise is deduced by Bayesian principle; (2) the LSSVR of G-L mixed homoscedastic noise (GLM−LSSVR) and G-L mixed heteroscedastic noise (GLMH−LSSVR) for complicate noise is developed; (3) the dual problem of GLM−LSSVR and GLMH−LSSVR is obtained using Lagrange-functional and according to KKT conditions; (4) the stability and effectiveness of the algorithm are guaranteed by solving GLM−LSSVR with the ALM method; and (5) the proposed technology is used to predict short-term wind speed by historical data, and then forecast the wind speed at some time after 10, 30, and 60 min, respectively. The comparison results display that the proposed model is better than classical technologies in statistical criteria.

In the same way, we can also study Gaussian–Laplacian, or Gaussian–Weibull mixed noise classification models. The new hybrid noise models would effectively solve complicated noise classification problems. 

## Figures and Tables

**Figure 1 entropy-22-00629-f001:**
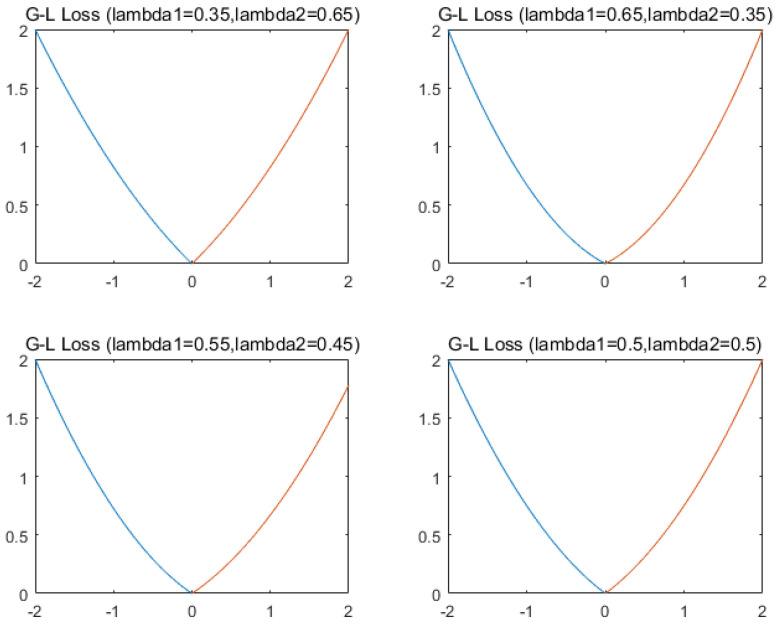
G-L empirical risk loss of different parameters.

**Figure 2 entropy-22-00629-f002:**
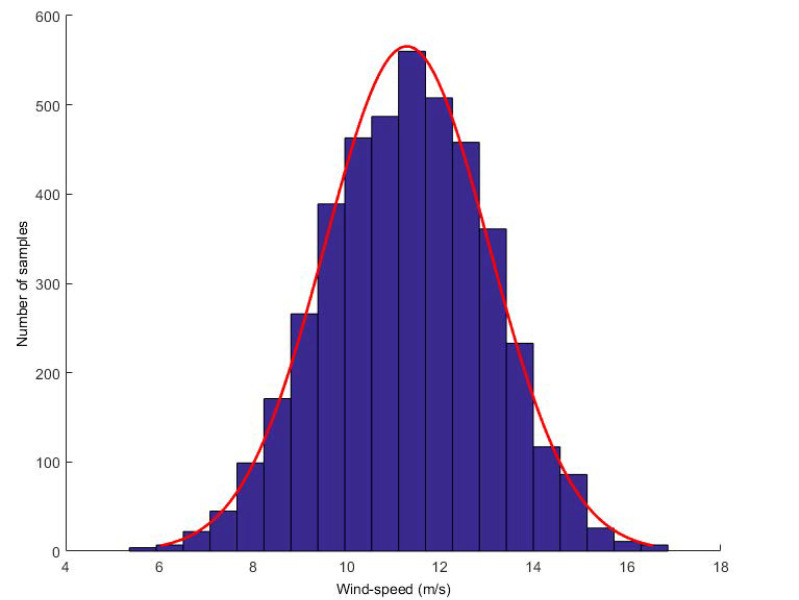
High wind speed distribution.

**Figure 3 entropy-22-00629-f003:**
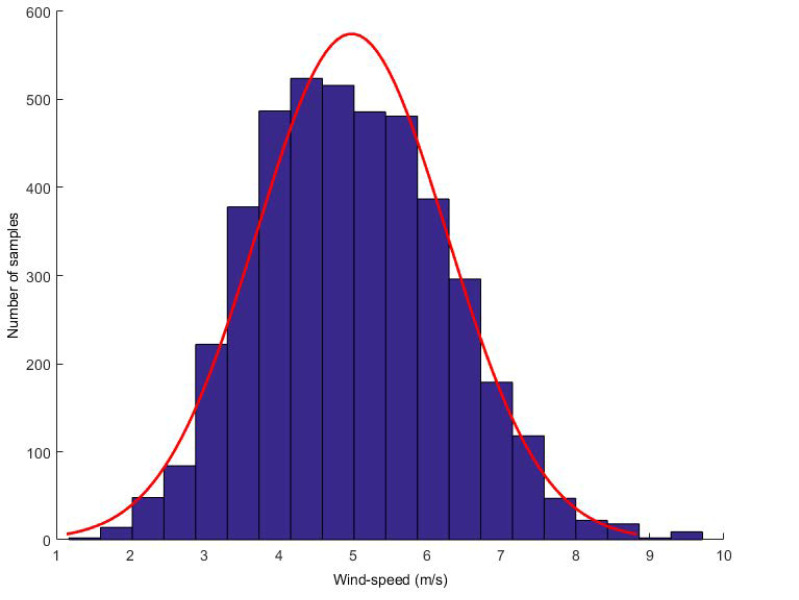
Low wind speed distribution.

**Figure 4 entropy-22-00629-f004:**
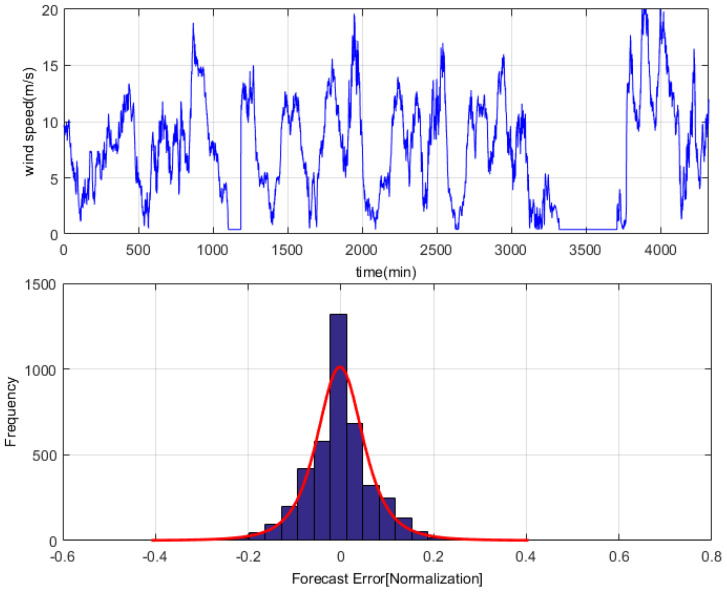
G-L mixed distribution of wind-speed forecasting-error with the persistence method.

**Figure 5 entropy-22-00629-f005:**
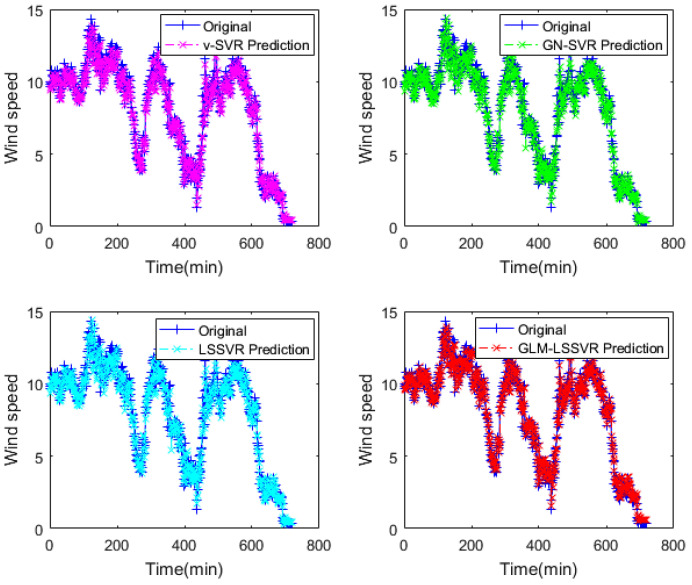
Result of four wind-speed forecasting models after 10 min.

**Figure 6 entropy-22-00629-f006:**
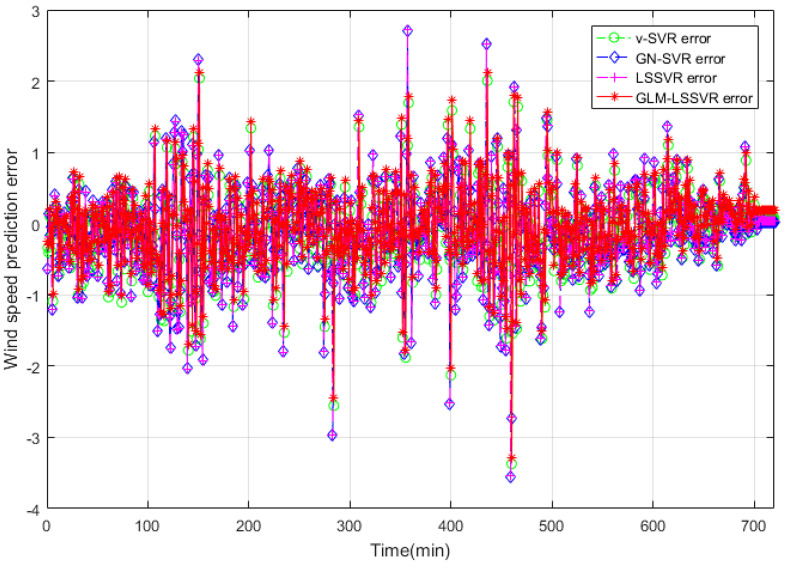
Error of four wind-speed forecasting models after 10 min.

**Figure 7 entropy-22-00629-f007:**
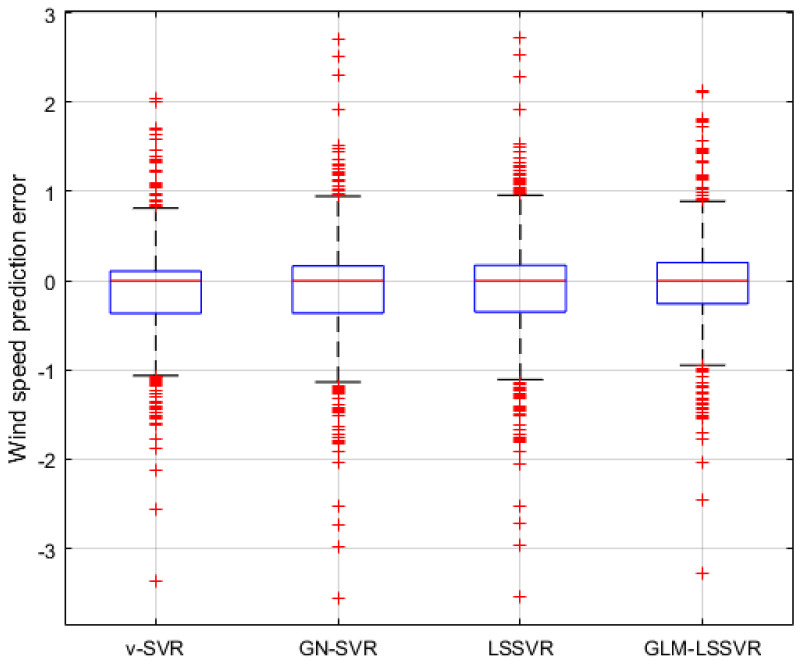
Residual box plot of four wind-speed forecasting models after 10 min.

**Figure 8 entropy-22-00629-f008:**
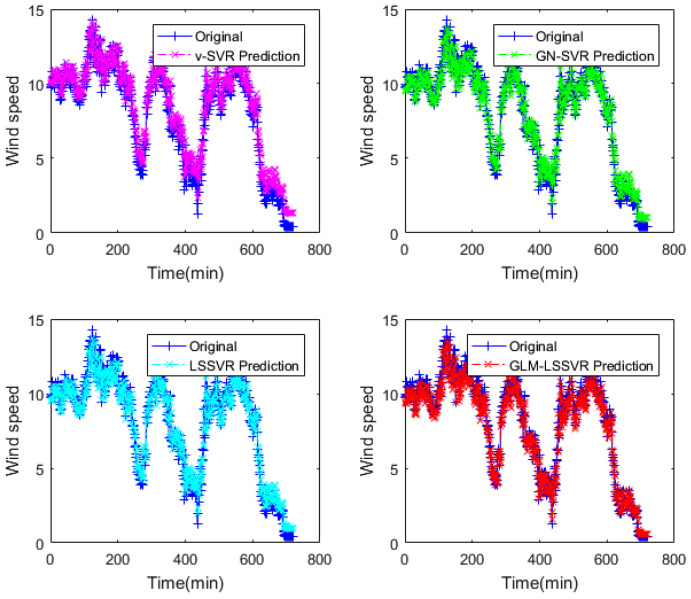
Result of four wind-speed forecasting models after 30 min.

**Figure 9 entropy-22-00629-f009:**
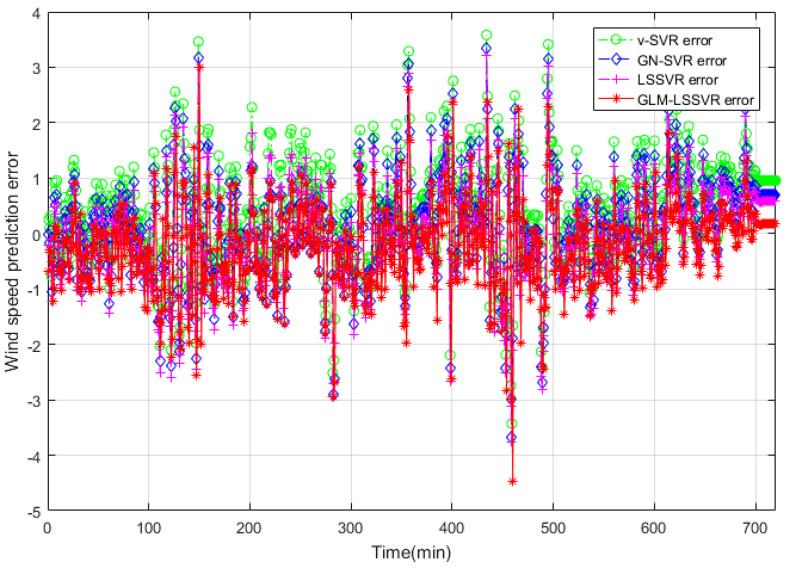
Error of four wind-speed forecasting models after 30 min.

**Figure 10 entropy-22-00629-f010:**
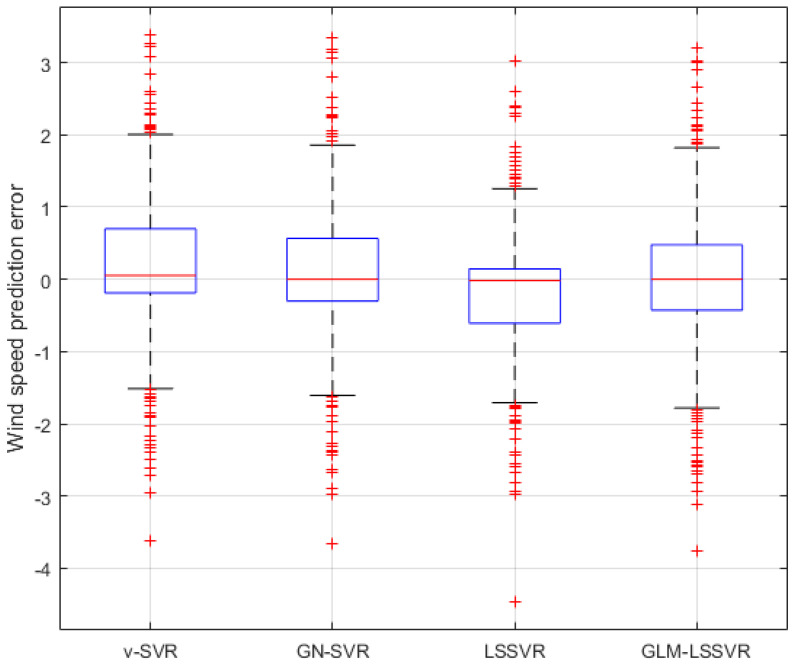
Residual box plot of four wind-speed forecasting models after 30 min.

**Figure 11 entropy-22-00629-f011:**
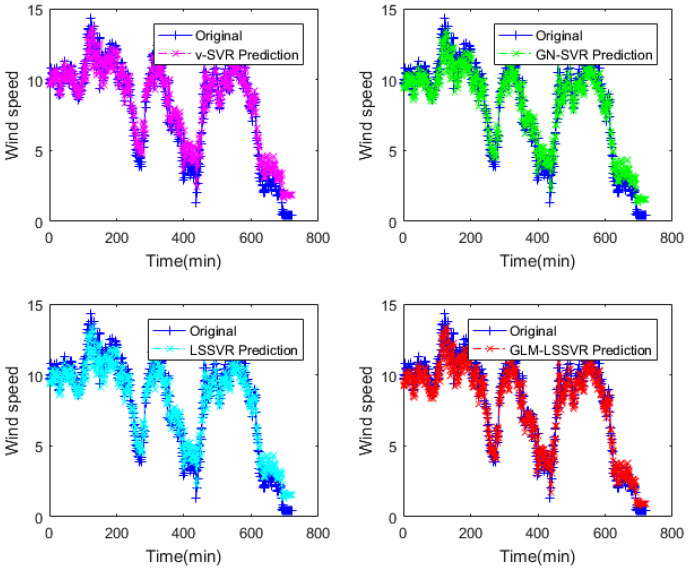
Result of four wind-speed forecasting models after 60 min.

**Figure 12 entropy-22-00629-f012:**
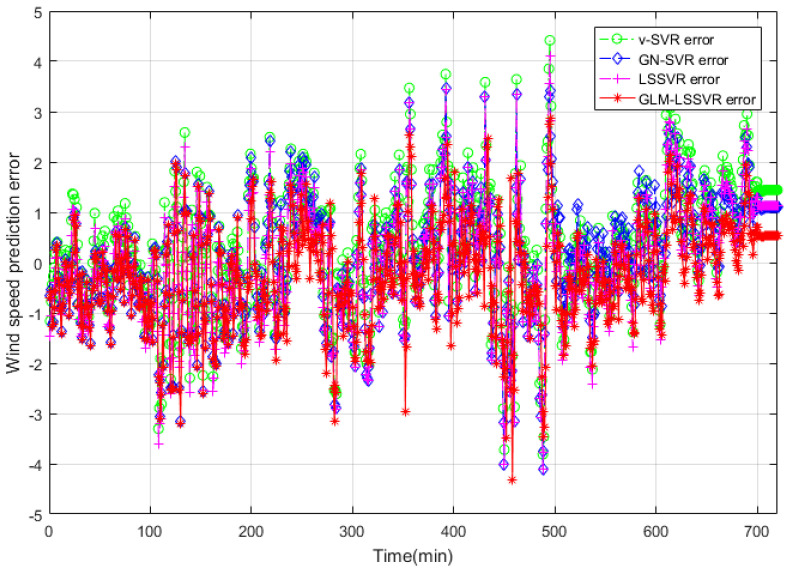
Error of four wind-speed forecasting models after 60 min.

**Figure 13 entropy-22-00629-f013:**
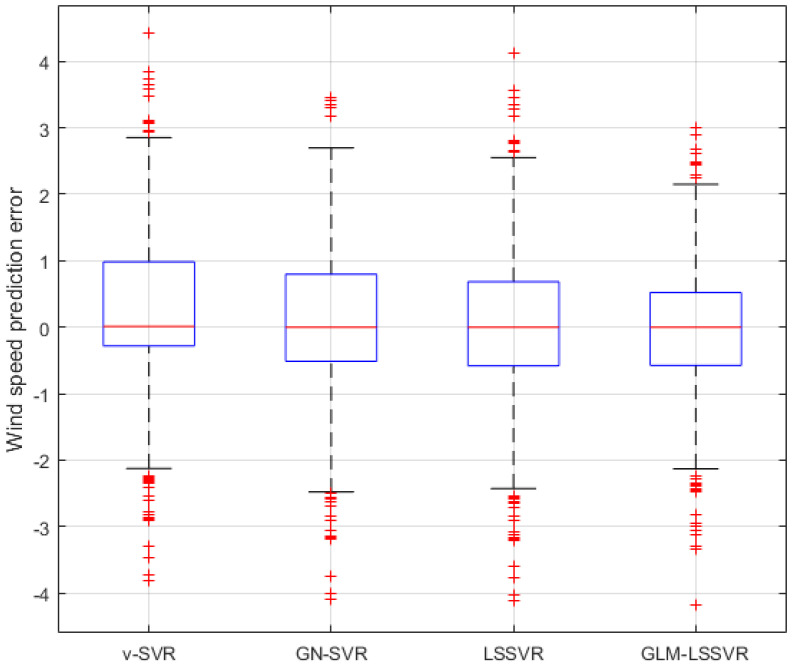
Residual box plot of four wind-speed forecasting models after 60 min.

**Table 1 entropy-22-00629-t001:** Error statistic of four wind-speed forecasting models after 10 min.

Model	MAE (m/s)	RMSE (m/s)	MAPE (%)	SEP (%)
ν−SVR	0.4280	0.5833	8.02	7.12
GN−SVR	0.4256	0.5789	7.92	7.07
LSSVR	0.4219	0.5768	7.94	7.06
GLM−LSSVR	0.4190	0.5711	7.91	7.05

**Table 2 entropy-22-00629-t002:** Error statistic of four wind-speed forecasting models after 30 min.

Model	MAE (m/s)	RMSE (m/s)	MAPE (%)	SEP (%)
ν−SVR	0.7979	1.0116	23.36	12.53
GN−SVR	0.7368	0.9886	19.93	11.89
LSSVR	0.7109	0.9226	17.17	11.43
GLM−LSSVR	0.6185	0.8241	10.71	10.19

**Table 3 entropy-22-00629-t003:** Error statistic of four wind-speed forecasting models after 60 min.

Model	MAE (m/s)	RMSE (m/s)	MAPE (%)	SEP (%)
ν−SVR	0.9994	1.2580	33.93	15.66
GN−SVR	0.9728	1.2355	31.78	15.37
LSSVR	0.9646	1.2177	29.01	15.16
GLM−LSSVR	0.8835	1.1180	25.72	13.97

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
