# Peer review of "LSSVR Model of G-L Mixed Noise-Characteristic with Its Applications"

_entropy, 2020, doi:10.3390/e22060629_

Round 1
Reviewer 1 Report
This article presents a new technique to forecast wind speed. Interesting results are found using the newly proposed method, while the current manuscript needs further modifications such as:
- The introduction lacks an overview of the existing techniques, could be found in the literature, that are used for wind speed forecasting including (i) mathematical-based methods, (ii) statistical-based methods, and (iii) AI-based method. I would suggest adding extensive literature on existing methods in these three areas before moving to section 2.
- Can the LSSVR algorithm work with the noise of different types of Gaussian? Please discuss this in details.
- Figure 2 has low quality, please modify accordingly. In addition, please specify the source of the data shown in Figure 2 and how this data has been measured, which sensor, accuracy, etc.
- Again, the data shown in Figure 3 and 4 are plotted without mentioning the accuracy of this data and what tools did you use to capture the wind speed. In addition, regarding the forecasted wind speed shown in Figure 4, I recommend adding the percentage of the error on the top of the graph (the same comment applies for Figure 7, 10 and 13).
- Can you compare the results shown in Figure 5 with 5-min or even lesser (say 1-min) resolution? this would typically help the readers to understand how effective is the implemented algorithm.
- Above-all what are the main limitations of the proposed technique? there is clearly detailed mathematical processing that needs to be done prior to the implementation of the algorithm, please outline the limitations for the algorithm and were possible have a flowchart describing the overall functionality and the error attained in each case study described in the manuscript (this ideally would be placed before the conclusion section).
Author Response
Response to Reviewer 1 Comments and Suggestions
Dear Reviewer 1
Thanks for your great help in bringing us constructive comments. It is very useful for us to improve the manuscript. We have carefully read the comments and revised the paper according to your suggestions. The detailed revisions are listed below.
Thanks again and best regards!
Sincerely
Shiguang Zhang, Ting zhou, Lin sun, Wei Wang, and Baofang Chang

Reviewer 2 Report
Reviewer comments
In this paper the authors a least square support vector regression-based model to forecasting the short-term wind speed. The abstract must be reformulated, as it is now the objectives of the work and the results are not understood. The sections of the models are full of mathematical formulations without an adequate introduction that justifies their treatment. In the case study section, there are many figures and little text explaining the results. The figures are not adequately described and analyzed. The results of the measurement metric are shown in Table 1 without an adequate explanation. Finally, the conclusions need to be reformulated and the possible practical uses of the technology and the future development of the work are lacking.
1-8) Reformulate the abstract, better explain what the reasons are for using this technology and explain the difference between single and multiple noise distributions.
12) It would be appropriate to start the introduction by dealing with the problem to which this technology is applied, therefore the wind speed. Once the problem is introduced, we can introduce the technology.
14-16) Explain the meaning of kernel-techniques
19-20) Adequately introduce the topic of noise distribution in the regression model before introducing references. In this way the reader can better follow the flow of information.
19-27) References to works are provided, but the link is not understood. it is necessary to follow a logical flow. What do we want to introduce? It is not clear.
28-29) Move this section to the beginning of the Introduction
28-36) Here the references are better, they are linked by a logical evolution, but it is necessary to better highlight the objective of the Introduction.
37-48) Good outline of the paper
52) check the format of equation (3), there seems to be more space
59-60) This paragraph is the copy of the previous one. Pease check.
60) Improve the caption of Figure 1. indicates the meaning of the parameters you enter (lamada ?). Maybe lambda
63) Move Figure 1 after its reference, the space is there
64-65) Define Mercer, and explain better the equation (6)
65-69) l parameter meaning is missing
70-76) Good introduction to the next section.
77-78) Explain better how you moved from equation (4) to (6). What causes the problem formulation to change? Explain the change in the empirical risk loss about Gauss-Laplace mixture homoscedastic noise
79) Adequately introduce the 2 theorems. Why are you using them? What will they help you with?
79-87) It is not clear why you introduced the two theorems and also presented the proof. Explain why it is important to report this mathematical treatment.
88-99) What is similar in this section, it is not clear why you are dealing with a mathematical formulation of theorem 3.
101-102) Finally explain what the two theorems are for, you have to move this reference first, so that the reader can understand the reason for this mathematical formulation.
109) Move Figure 2 after its reference, the space is there
165-167) These are 8 figures, you should explain them in detail or insert adequate explanations in the captions, otherwise the work of analyzing the results is left to the reader.
174-185) Conclusions need to be reformulated: first of all it is appropriate to move the figures so that the result set is grouped. As it is now it is difficult to read. Furthermore, it is necessary to summarize what has been done and what the results are, perhaps by making a comparison between the results that demonstrate the improvements obtained with the proposed method. Finally, the possible practical uses of the technology and the future development of the work are lacking.
Author Response
Response to Reviewer 2 Comments and Suggestions
Dear Reviewer 2
Thanks for your great help in bringing us constructive comments. It is very useful for us to improve the manuscript. We have carefully read the comments and revised the paper according to your suggestions. The detailed revisions are listed below.
Thanks again and best regards!
Sincerely
Shiguang Zhang, Ting zhou, Lin sun, Wei Wang , and Baofang Chang

Round 2
Reviewer 2 Report
accept
Author Response
Dear Reviewer 2
Thanks for your great help in bringing us constructive comments. It is very useful for us to improve the manuscript. We have carefully read the comments and revised the paper according to your suggestions. The detailed revisions are listed below.
Thanks again and best regards!
Sincerely
Shiguang Zhang, Ting zhou, Lin sun, Wei Wang , and Baofang Chang
Response : Thank you very much for your valuable suggestion.
Based on your suggestion, We will carefully improve the research design, adequately describe the method, and clearly present the results.

This manuscript is a resubmission of an earlier submission. The following is a list of the peer review reports and author responses from that submission.